# Piston Compression Ring Elastodynamics and Ring–Liner Elastohydrodynamic Lubrication Correlation Analysis

**Swagatika Biswal and Prakash Chandra Mishra \*** 

Department of Mechanical Engineering, Veer Surendra Sai University of Technology, Burla 768018, India
\* Correspondence: pcmishra_me@vssut.ac.in; Tel.: +91-8917-5354-45

**Abstract:** Friction loss in an internal combustion engine largely depends on elastohydrodynamic lubrication. The piston compression ring is a contributor to such parasitic losses in the piston subsystem. The complex elastodynamics of the ring are responsible for the transient and regime-altering film that affects the elastohydrodynamic lubrication of the ring liner contact conjunction. The current paper will discuss the ring radial, lateral deformation, and axial twist, and its effect on the film profile of the compression ring and its subsequent effect on tribological characteristics like elastohydrodynamic pressure, friction, and lubricant. A finite difference technique is used to solve the elastohydrodynamic issue of elastodynamic piston compression by introducing the elastodynamically influenced film thickness into the lubrication model. The results show that consideration of the elastodynamics predicts a 23.53% reduction in friction power loss in the power stroke due to the elastodynamic ring compared to the rigid ring. The elastodynamic effect improves the lubricant oil flow into the conjunction. A finite element simulation predicts a von-Mises stress of 0.414 N/mm$^2$, and a maximum deformation of 0.513 μm at the core and coating interface is observed at the ring–ring groove contact. The sustainability of EHL in this case largely depends on the ring–liner elastodynamics.

**Keywords:** compression ring; elastodynamics; ring–liner oil film thickness; fluid friction power; lubricant oil flow; sustainable elastohydrodynamic lubrication

## 1. Introduction

The conformed engine piston top compression ring sits on the top ring groove and is subjected to various modal deformation during the operating cycle of an IC engine. Guided by the cylinder bore wall/liner, it elastically flexes its shape due to the combined action of gas pressure, ring tension, lubricant reaction, and oil friction during the reciprocation of the piston assembly, engaged to generate mechanical power. The dynamic ring governs the transient film shape responsible for the hydrodynamic action of the lubricant, due to which simultaneous sealing and sliding actions of the ring ensure durable piston assembly and engine performance. The hydrodynamic action in such a case is elastohydrodynamic in nature, not out of elastic deformation of the contiguous part due to extremely high concentrated contact as occurs in ball bearings, but because of the global deformation of the flexible compression ring (incompletely conformed to the cylinder bore). Therefore, there is a strong correlation between the elastodynamics of the compression ring and the elastohydrodynamic lubrication of the ring–liner conjunction. To explore such an interrelationship, it is required to understand the ring elastic deformation and how it replicates such a mode while in conformance with the piston–cylinder subsystem.

The dynamics of an incomplete ring were first studied by Lamb [1], who considered the analysis of a bar of circular arc shape for the in-plane flexural strength. The vibration of such free–free curved was discussed considering the center as the origin and by deriving the motion equations in polar form. Subsequently, Den Hartog [2] derived the first and second natural frequencies of an incomplete ring which was either fixed or allowed to rotate

at the ends. The non-extensional vibration led to the observation of in-plane deformation in different modes. Brown [3], considering such modal behavior of a ring-shaped bar, studied out-of-plane combined torsional and flexural vibration perpendicular to the ring plane using a modified Rayleigh's method. Archer [4] worked on the in-plane extensional vibration of the incomplete circular ring by clamping it at one end and observing the time-dependent deformation at the other end using the basic equation of motion proposed by Love [5]. Volterra and Morell [6] investigated the out-of-plane vibration of an elastic arc clamped at both ends for the lowest natural frequency.

Rao [7] investigated the influence of transverse shear and inertia of rotation on a circular ring subjected to combined twisting and bending. The governing equation in this case is prepared from a modified Hamilton's equation including rotary inertia and deformation due to transverse shear. In this analysis, the vibration characteristics are studied for an incomplete circular ring with a rectangular cross-section. Endo [8] formulated the equation to find out the out-of-plane vibration of an incomplete ring at a random cross-section using Ritz's method. Hawking [9] carried out a mathematical formulation of in-extensional vibration of an unrestricted incomplete circular ring with a uniform cross-section, which can predict flexural, torsional, and shear waves.

However, the analyses [1–9] are limited to free rings, which are unlike the piston ring conformed to the cylinder bore, and in most cases, the outcome is natural frequency and many times the ring is not incomplete. Such ring dynamic analysis could not correlate the ring's global deformation to the instantaneous film profile of ring–liner conjunction. Therefore, it remains isolated from the lubrication aspect of the piston ring-cylinder liner co-action. The elastohydrodynamic lubrication of piston ring was first introduced by Dowson et al. [10], and it was considered a self-adjusted mechanism of simultaneous sealing and sliding due to piston reciprocation and variable combustion gas pressure. This mechanism flexes the ring to form a mixed/boundary regime near to the dead center which is hydrodynamic during the resting part of the engine cycle. Furuhama and Sasaki [11] carried out an experimental investigation to trace the cyclic variation of ring–liner friction with a technique involving a floating liner and a pressure-balancing device. The test was conducted in small-size diesel as well as gasoline engines operated with multi-grade lubricant oil and friction modifiers. Here, the ring dynamics effect is not traceable in these studies [10,11].

Tian et al. [12–16] modeled the piston ring dynamics to investigate the twist and associated gas blow-by due to the axial flutter of the ring within the piston groove lands. Piston ring motion and combustion gas flow were coupled to investigate the ring flutter and associate blow-by linked to the positive and negative twists [14,15] of piston rings. Liu et al. [13,16] investigated the transient dynamic motion of the engine piston subsystem and its effect on the transient friction phenomena in the engine cycle. A 2D averaged Reynolds equation was solved to trace the cyclic response of fluid friction and power for the piston subsystem and the ring pack were compared. Rahmani et al. [17] investigated a partially conformed multi-lobed ring-bore conjunction in the transient regime of lubrication from hydrodynamic in mid-stroke to mixed at the region of significant pressure build-up and boundary near to the top and bottom dead centers for the tribological performance considering new as well as worn piston rings.

Baker et al. [18–20] discussed the elastodynamics of piston compression rings considering ring in-plane and out-of-plane dynamics coupled with the ring–liner lubrication. The studied this along with a 1D combustion gas flow technique considered to understand the blow-by characteristics. Dlugoš and Novotný [19] formulated a 3D piston ring technique implementing a multi-body system of Timoshenko beam theory and compared it with a finite element model. Through this approach, a model is developed to deal with the distorted bore shape and incomplete piston ring geometry, including ring gap, to study deflection and motion, such as the ring twist effect on an unsymmetrical ring section. Liu et al. [21–23] investigated a piston ring pack to simulate its structural dynamics coupled with gas pressure dynamics and oil transport.

Baker et al. [23] investigated the 3D tribodynamic technique to discuss the coupled ring elastodynamics and discussed the globally deformed ring profile at various crank angles along with the cyclic variation of power loss for rings, comparing the rings subjected to rigid dynamics, in-plane dynamics, and complete dynamics. There are studies reported on compression ring sweep excitation [24] and cylinder deactivation [25] on the lubrication characteristics of the ring–liner conjunction. The kinematics and dynamics of the ring simulated using CFD showed a fluid-structure analysis [26], and the influence of compression ring [26] dynamics is always highlighted in the literature in tribological characteristics of ring–liner contact. Many experimental techniques have been developed to study the influence of coating in reducing friction losses in IC engines [27]. Furthermore, friction losses in the piston–cylinder subsystem were investigated in the particular case of isochoric pressure gain [28], including that of an F1 racing car engine [29]. Studies revealed [30,31] that inlet air pollutants play an important role in engine component wear, as they mix with engine oil and pass through the ring–liner conjunction and cause wear due to abrasive dust [32]. Inclusion of contaminants in lubricant oil and its quantification will be a way forward in engine tribology research [33–35] for developing green lubricants [36].

Based on this broad survey of the literature, it is confirmed that the compression ring elastodynamics are not like a free ring fluttering, but constrained in-ring in the groove land and guided by the cylinder liner involving fluid film lubrication, mixed regime, and boundary regime at different locations of piston motion. It is fully interlinked to the gas dynamics, elastohydrodynamics, and asperity interaction-like phenomena, where deformed film shape is a governing parameter in film shape formation as well as subsequent performance modification. Hence, correlation analysis, which is important for EHL sustainability, is the main objective of this research work.

## 2. Theories of Ring Elastodynamics

### 2.1. In Situ Ring Elastodynamics

Figure 1 shows the axis consideration for ring 3D deformation. The axis u is in the ring axial direction (out-of-plane), axis v is in the ring tangential direction (in-plane) and axis w is the in-plane radial direction. All these axes are mutually perpendicular to each other. Figure 2a–c shows the details of forces that may act on the compression in three different crank positions. The forces acting on the elastic compression ring are resolved into two components: one is the net radial force responsible for in-plane deformations and the second is net axial force responsible for out-of-plane deformations.

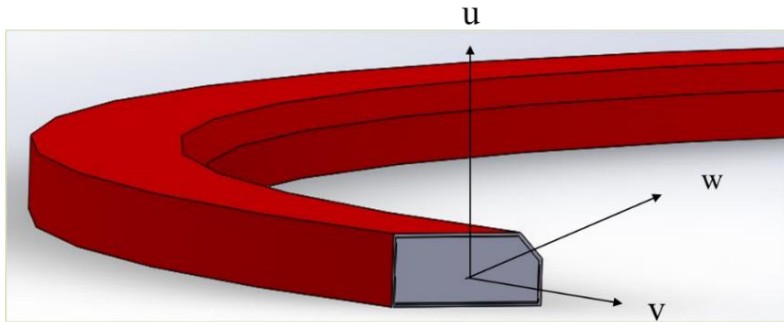

**Figure 1.** Axis consideration of ring elastodynamics behavior.

The net in-plane force $F_{in-plane}(\varphi, t)$ is given in Equation (1) as per Baker et al. [23].

$$(F_e + F_g) - (W_h + W_a) = F_{in-plane} \tag{1}$$

here, $F_e$ is the ring elastic force acting radially outward once conformed in the cylinder bore. It is constant force throughout the engine cycle for a particular ring material, with an assumption that the ring material does not degrade during the engine operation. $F_g$ is

the gas pressure force working along the faces of the compression ring, including the back face parallel to the axial direction and slant face, as well as the face parallel to in-plane. $W_h$ is the force generated due to the hydrodynamic action of the lubricant oil, and it always has the tendency to keep ring–liner separation and acts radially inward. Other than $F_e$, all contributory forces to the in-plane mode show cyclic variation in the engine cycle. For example, combustion pressure builds up in the compression-power stroke transition, at (300–400°) crank location. Whereas the asperity contact reaction, $W_a$, is significant near to the dead centers (TDC/BDC) due to instant seizure of ring sliding for motion reversal [26], though the squeeze film action, even if negligible, is still active. The groove–land sliding friction is neglected here due to a comparatively lower magnitude to other forces [22].

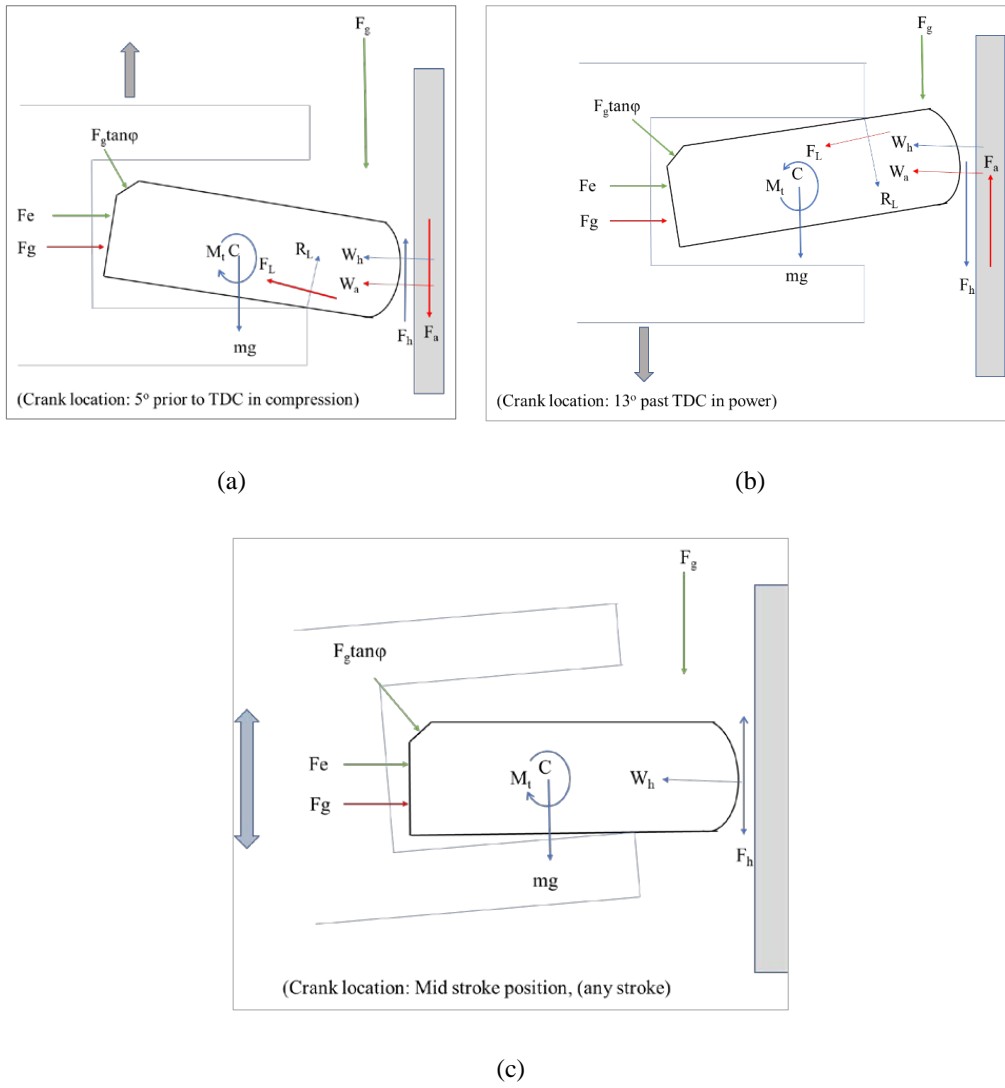

(a)

(b)

(c)

**Figure 2.** In situ piston compression ring elastodynamics free body diagram: (**a**) at crank location 5° prior to TDC on compression, (**b**) at crank location 13° past TDC on power, (**c**) mid-stroke position (any stroke).

Furthermore, the equation of dynamic motion that describes the in-plane ring response was developed by Lang [27] and is presented in Equations (2) and (3).

$$\frac{\partial^6 v}{\partial \varphi^6} + 2\frac{\partial^4 v}{\partial \varphi^4} + \frac{\partial^2 v}{\partial \varphi^2} + \frac{1}{p\omega_0^2}\frac{\partial^4 v}{\partial \varphi^2 \partial t^2} - \frac{1}{p\omega_0^2}\frac{\partial^2 v}{\partial t^2} = \frac{R^3}{EI}\left\{ \frac{\partial^2}{\partial \varphi^2}\left[ F_{in-plane}(\varphi, t) \right] - [F_T(\varphi, t)] \right\} \tag{2}$$

$$\frac{\partial^6 \omega}{\partial \varphi^6} + 2\frac{\partial^4 \omega}{\partial \varphi^4} + \frac{\partial^2 \omega}{\partial \varphi^2} + \frac{1}{p\omega_0^2}\frac{\partial^4 \omega}{\partial \varphi^2 \partial t^2} - \frac{1}{p\omega_0^2}\frac{\partial^2 \omega}{\partial t^2} = \frac{R^3}{EI}\left\{ \frac{\partial^2}{\partial \varphi^2}\left[ F_{in\_plane}(\varphi, t) \right] - [F_T(\varphi, t)] \right\} \tag{3}$$

where, $p = \frac{1}{AR^2}$, $\omega_0 = \frac{E}{\rho R^2}$, and $F_T$ (the shear force acting in the tangential direction) are neglected here for their small magnitude compared to other applied forces. The output of the equation motion solution helps in calculating the ring in-plane modal shape, as a function of the net force and time interval, $t$. The estimated deformation is sent to the lubrication model to update the ring–liner gap geometry.

### 2.2. Out-of-Plane Dynamics of the Piston Compression Ring

The force acting in the out-of-plane direction in a compression ring is given in Equation (4):

$$\left( mg + F_g \right) - \left( F_h + F_a + R_g \right) = F_{out\_plane} \tag{4}$$

here the weight of the ring ($mg$) is constant throughout the engine cycle considering the wear rate is very slow, and the ring lasts more than 100,000 km in a commercial engine [26]. Other out-of-plane contributing forces are $F_g$, the combustion gas force; $F_h$, the force of hydrodynamic friction; and the groove reaction, $R_g$, due to ring sliding in contact with groove land.

The equation of motion for the out-of-plane mode takes a parallel form to the in-plane motion and is expressed as out-of-plane displacement, $u$, and the ring twist, $\beta$, which is given in Equations (5) and (6).

$$\frac{\partial^4 u}{\partial \varphi^4} - R\frac{\partial^\beta 2}{\partial^2 \varphi} - k\left( \frac{\partial^2 u}{\partial \varphi^2} - R\frac{\partial^2 \beta}{\partial \varphi^2} \right) = \frac{mR^4}{EI_x}\left( \frac{\partial^4 u}{\partial t^4} - \frac{F_{out\_of\_plane}}{m} \right) \tag{5}$$

$$\frac{\partial^2 u}{\partial \varphi^2} - R\beta + k\left( \frac{\partial^2 u}{\partial \varphi^2} + R\frac{\partial^2 \beta}{\partial \varphi^2} \right) = \frac{mR^4}{EI_x}\left( \frac{\Phi}{m} \right) \tag{6}$$

here $\Phi$ is the torsional loading. The eigen value equation for this above relation is given in Equation (7).

$$\frac{\partial^4 \beta_n}{\partial \varphi^4} + 2\frac{\partial^2 \beta_n}{\partial \varphi^2} + \beta_n = -\lambda_n\left( \frac{1+k}{R} \right)u_n \tag{7}$$

The set of uncoupled equations is given as Equation (8).

$$\overset{**}{\xi}_n + \omega_n^2 \xi_n = \frac{Q_n}{m} f(t) \tag{8}$$

The excitation force (given in Equation (9)) thus derived will be included in ring dynamic solution.

$$Q_n = \frac{\int_0^\alpha \left( F_{out\_of\_plane}(\varphi)u_n \right)d\varphi}{\int_0^\alpha u_n^2 d\varphi} \tag{9}$$

The solution of Equation (5) requires the solution of Equations (6) and (7) which will take the form of Equation (10).

$$u(\varphi, t) = \sum_{n=1}^\infty U_n(\varphi)\xi_n(t) \tag{10}$$

The ring liner conjunction and tribological performance are greatly influenced by the elastodynamic modal behavior of the ring. The reason for this is that the resultant force on

the ring–liner conjunction is crank position specific. Considering the next crank angle as the subsequent step of increment of angle, the resultant ring circumferential shape alters the film thickness to decide the elastohydrodynamic performance. For the subsequent elastodynamic model, updated ring deformation and sliding velocity are taken as the initial conditions and once again the elastohydrodynamic lubrication performance is evaluated, and so on, to trace the transient response of ring–liner contact. The elastodynamic modes such as in-plane, out-of-plane, and ring twist are included in order to detect the extreme ring modal deformation at any crank location.

### 2.3. Film profile of Piston Compression Ring and cylinder liner conjuction

The piston ring and cylinder–liner conjunction film thickness is derived in Equation (11). $h_0$ is the minimum film between the flexible compression ring and the cylinder bore surface [19,20]. The geometry function is considered in parabolic form as a single point minima-type parabolic profile [2]. $\Delta_{i,j}$ is the global deformation of the compression ring out of inward and outward springing motion because of combined action of various forces as per Figure 2a–c. Furthermore, $\delta_{i,j}$ is the local elastic deformation, which is negligible here due to its exceptionally low order [0.1 nm to 10 nm] [2].

$$h = h_0 + S_{i,j} + \Delta_{i,j} + \delta_{i,j} \tag{11}$$

Due to this consideration, film thickness becomes specific and is reduced to

$$h = h_0 + S_{i,j} + \Delta_{i,j} \tag{12}$$

In the tribological simulation of the piston–ring cylinder conjunction, a fully flooded inlet boundary condition is taken for evaluation. The oil temperature is kept constant for the entire engine. Here, the temperature of the cylinder liner is more comparable to any generated temperature out of viscous shear heating in a short interval of time along a small width of contact, as demonstrated by Morris [25].

### 2.4. Lubricant Rheology of Elastodynamic Ring and Liner Conjunction

The elastohydrodynamic lubrication and the associated pressure are largely affected by the lubricant rheology [27]. Properties such as viscosity, density, pressure, and temperature interrelationship play an important role in this alteration [28]. The dynamic viscosity is the game changer to be evaluated for each crank position for the ring–liner contact conjunction for the total engine cycle [20]. Mishra et al. [37] used the piezo-viscous model to address the pressure-viscosity interdependency. Further, the viscosity–temperature–pressure and density–pressure–temperature interrelationships are discussed in detail in Equation (13) and Equation (15), respectively.

### 2.5. Viscosity–Temperature–Pressure Interrelation

The effect of temperature and pressure variation on lubricant viscosity is given in Equation (13).

$$\eta = \eta_0 \exp\left\{ (\ln \eta_0 + 9.67) \left[ \left( \frac{\Theta - 138}{\Theta_0 - 138} \right)^{-S_0} \left( 1 + \frac{p - p_{atm}}{1.98 \times 10^8} \right)^Z - 1 \right] \right\} \tag{13}$$

where $\Theta = \Theta + 273$ and $\Theta_0 = \Theta_0 + 273$, and
here, Z and $S_0$ are lubricant specific parameters

$$Z = \frac{\alpha_0}{5.1 \times 10^{-9}[\ln \eta_0 + 9.67]} \quad and \quad S_0 = \frac{\beta_0(\Theta_0 - 138)}{\ln \eta_0 + 9.67} \tag{14}$$

where, $\alpha_0$ *and* $\beta_0$ are the piezo-viscous and thermo-viscous coefficients [38–40], respectively.

### 2.6. Density–Temperature–Pressure Interrelation

Here, due to weak hydrodynamic pressure, the density remains unaltered ($\rho_0 = 833.8$ kg/m$^3$), which is evidenced from trials of taking the highest oil temperature and pressure in the ring–liner conjunction. The variation of lubricant bulk rheology (density, viscosity) is:

$$\rho = \rho_0 \left(1 - 0.65 \times 10^{-3} \Delta\theta\right) \left[1 + \frac{6 \times 10^{-10}(p - P_{atm})}{1 + 1.7 \times 10^{-9}(p - P_{atm})}\right] \tag{15}$$

Again, intermittent viscosity is also affected due to the oil shearing rate, along with changes in the level of lubricant pressure and temperature [40]. Hence, shear rate needs to be estimated. In addition, due to faster moving reciprocation, the second Newtonian viscosity is more influential than the first during a high shear rate in the engine warm-up period, excluding close crank position to the dead centres. In this circumstance, the shear-dependent viscosity is given in Equation (16) and is known as the Cross equation [40].

$$\mu = \mu_2 + \frac{(\mu_1 - \mu_2)}{1 + \beta(\gamma*)^k} \tag{16}$$

where $\beta$ *and* $k$ are lubricant-dependent parameters for fitting and $\gamma* = \frac{|u|}{h_T}$.

The viscosity ratio considering the primary and secondary shear is presented in Equation (17)

$$\mu_r = \frac{\mu_1 - \mu_2}{\mu - \mu_2} = 1 + \beta(\gamma^*)^\kappa \tag{17}$$

Tables 1–3 present the input parameters in piston–liner contact analysis.

**Table 1.** Ring elastodynamic parameters of the compression ring.

| Ring Elastodynamic Parameter | Value |
| --- | --- |
| Elastic modulus | 205 GPa |
| Poisson's ratio | 0.27 |
| Ring density | 7.850 g/mm$^3$ |
| Ring radial width | 3.2 mm |
| Ring axial width | 1.2 mm |
| Nominal radius of fitted ring | 44.52 mm |
| Ring second moment area | $2.25 \times 10^{-12}$ mm$^4$ |

**Table 2.** Ring roughness parameters of the compression ring.

| Ring Roughness Parameter | Value |
| --- | --- |
| Ra for the liner | 0.26 μm |
| Ra for the ring | 0.408 μm |
| Roughness parameter $(\zeta\kappa\sigma_c)_c$ | 0.074 |
| Measure of asperity gradient $(\sigma_c/\kappa)_c$ | 0.309 |

**Table 3.** Lubricating oil parameters for the compression ring–liner conjunction.

| Lubrication Parameter | Value |
| --- | --- |
| Pressure–viscosity coefficient | $2 \times 10^{-8}$ m$^2$/N |
| Thermal expansion coefficient | $6.5 \times 10^{-4}$ 1/K |
| Lubricant oil density | 833.8 at 40 °C, 783.8 at 100 °C |
| Lubricant kinetic viscosity | 59.99 at 40 °C, 9.59.8 at 100 °C |

### 2.7. Theory of Ring Elastohydrodynamics

Further assumptions on the lubrication of piston ring cylinder contact with zero side leakage on ring circumferential direction as entrainment of lubricant in the axial direction of ring–liner contact

$$\frac{\partial}{\partial x}\left(\frac{\rho h^3}{\eta}\frac{\partial p}{\partial x}\right) + \frac{\partial}{\partial y}\left(\frac{\rho h^3}{\eta}\frac{\partial p}{\partial y}\right) = 12\left(u_{av}\frac{\partial}{\partial x}(\rho h) + \frac{\partial}{\partial t}(\rho h)\right) \tag{18}$$

here, pressure density and viscosity are considered unchanged across the film profile, while solving the above Reynolds equation derived from the Navier stokes equation. To reach such an equation, the inertia term is neglected considering the tiny mass of the lubricant flowing into the conjunction.

The Reynolds equation in this case is solved using the finite difference method (FDM) on two-dimensional forms. Here, the modally deformed ring profile-induced film thickness is simultaneously solved with the governing equation.

In most parts of the engine cycle, other than the dead centers or the vicinities, the contact friction of the piston subsystem, especially the piston ring, happens due to viscous action out of the hydrodynamic/elastohydrodynamic mechanism. In the vicinity of the dead centers, due to instant reversal, the motion ceases and leads to the hydrodynamic action only remaining due to squeeze-film action, and it is comparatively less compared to the asperity contact pressure/friction. Hence, the generic equation of applied force balance needs to be crank location specific, as shown in Figure 2a–c.

For cavitation inclusion, the Swift–Stieber boundary condition was implemented, which takes any negative pressure formation in conjunction with the cavitation zone and sets the pressure gradient at zero in the lubricant film rupture zone [37]. The minimum film thickness, hm, at a particular time step has angle of twist, β. The compression ring profile affects its axial contact profile with the surface of the bore. For a tilt angle β (evaluated through out-of-plane ring dynamics analysis), a simplified coordinate transfer is considered: $s(x, y) = x \sin \beta + s(x) \cos \beta$.

An initial assumption of the film is considered, and the computation is carried out using an iterative process with an over-relaxation coefficient. The computed pressure profile is obtained.

$$p_{i,j}^n = (1-\gamma)p_{i,j}^0 + \gamma p_{i,j}^n \qquad (0 \prec \gamma \prec 2) \tag{19}$$

The pressure relaxation factor here is case specific. The convergence of hydrodynamic pressure at each node is evaluated by using the following criteria (20):

$$Error_{pressure} = \frac{\sum\limits_{i=1}^{I}\sum\limits_{j=1}^{J}\left|p_{i,j}^n - p_{i,j}^n\right|}{\sum\limits_{i=1}^{I}\sum\limits_{j=1}^{J}p_{i,j}^n} \leq 1 \times 10^{-5} \tag{20}$$

By integrating this computed hydrodynamic pressure over the entire bearing area, the load bearing ability $W_h$ is evaluated:

$$W_h = \int\int p\,dA \tag{21}$$

Due to thinner lubricant films, there is a higher probability of one-to-one interactions of asperities on the ring–liner interface. Hence, in considerable parts of the engine cycle, a mixed regime of lubrication is always possible. For a predefined Gaussian distribution of asperities, Greenwood and Tripp [41] developed a model for the calculation of the asperity contact force in the boundary regime of lubrication, which is also for coated surface contact friction [42].

$$W_a = \frac{16\sqrt{2}}{15}\pi(\zeta\kappa\sigma)^2\sqrt{\frac{\sigma}{\kappa}}E'AF_{5/2}(\lambda_s) \tag{22}$$

where the parameters $(\zeta\kappa\sigma)$ and $(\sigma/\kappa)$ are dimensionless surface roughness parameters, acquired through rough surface measurement. $\kappa$ is the average radius of curvature of a single asperity, $\zeta$ is the number of asperity peaks available per unit area, and $\sigma$ denotes the composite rough surface parameter. 'A' is the apparent surface contact area (assuming that the ring face is smooth before adding roughness). $\lambda_s$ is the Striebeck film ratio; the ratio of film thickness to average surface roughness. $F_{5/2}(\lambda_s)$ is the probability distribution function of asperity height. In this analysis, it is approximated by a fifth-order polynomial curve fit.

$$F_{5/2}(\lambda) = -0.0046\lambda_s^5 + 0.0574\lambda_s^4 - 0.2958\lambda_s^3 + 0.7844\lambda_s^2 - 1.0776\lambda_s + 0.6167 \qquad (23)$$

The total friction of the ring–liner contact is the summation of friction due to viscous shear and that of boundary contact, which is given in Equation (24).

$$f_t = f_v + f_b \qquad (24)$$

Furthermore, boundary friction is the sum of the forces due to limiting Eyring stress and that of boundary shear. Styles et al. [43] used the formulation for the boundary as per Equation (25).

$$f_b = \tau_0 A_e + \xi W_a \qquad (25)$$

where $\tau_0$ is the limited Eyring stress and $\xi$ is the coefficient of pressure for boundary shear of surface asperities. Styles et al. [43] monitored such coefficients for the coatings of the ring face in contact, which is extracted for use in this study. The values were 0.3038 and 0.2012 for a new and worn ring, respectively. $A_e$ is the real area of contact for asperity, calculated by the summation of the contact area at the tip of the asperities.

$$A_e = \pi^2(\zeta\kappa\sigma)^2 A F_2(\lambda_5) \qquad (26)$$

Furthermore, the viscous component of overall friction is given in Equation (27):

$$f_b = \tau(A - A_e) \qquad (27)$$

The fluid shear stress is given in Equation (28):

$$\tau = \left(\tau_x^2 + \tau_y^2\right)^{\frac{1}{2}} = \left|\pm\frac{h}{2}\Delta p + V\frac{\eta}{h}\right| \qquad (28)$$

*2.8. Assumption of the Combined Elastodynamic and Elastohydrodynamic Analysis*

For the ring in-plane dynamics, the compression ring is treated as a thin wall structure due to its nominal diameter in ratio with radial thickness yielding a value greater than 12 mm. There is no rotary inertia in the ring, that means the motion is dominated only through in-plane and out-of-plane motion. The free–free boundary condition for deformation calculation was considered. The pressure and temperature for the upper as well as lower boundary are assumed. No leakage of lubricant in the ring circumferential direction other than that along the thin ring gap of the conformed ring are in order of 0.1–0.2 mm (Mishra et al. [37]). Friction between the groove and ring is negligible compared to other active forces. The gas pressure along the ring gap is the average of the pressure crank case and the cylinder pressure. Further boundary conditions are listed in Equations (29)–(36).

$$p(0, y) = p_l \qquad (29)$$

$$T(0, y) = T_l \qquad (30)$$

$$p(b, y) = p_t \qquad (31)$$

$$T(b, y) = T_t \qquad (32)$$

$$p(x_c, y) = p_c \tag{33}$$

$$T(x_c, y) = T_c \tag{34}$$

$$p(x, 0) = p(x, 2\pi) = \frac{p_l + p_t}{2} \tag{35}$$

$$T(x, 0) = T(x, 2\pi) = \frac{T_l + T_t}{2} \tag{36}$$

### 2.9. Validation of the Model

The current simulation results of the lubrication model were compared with the experimental friction of Furuhama and Sasaki [11] and are presented in Figure 3. The deviation in suction stroke is 8.34% less compared to the referred work, while it is 9.4% in compression, 2.7% in power, and 15.3% in exhaust. The variation may be due to the complex heat transfer across the wide range of material in the engine structure.

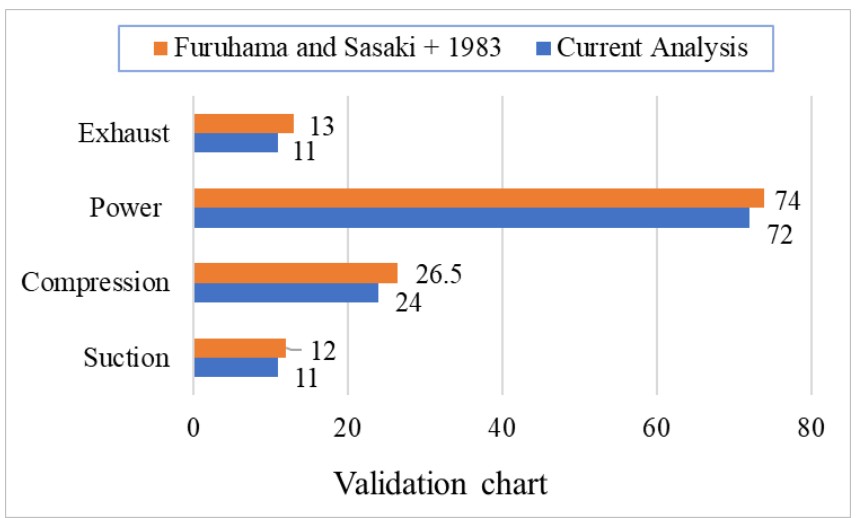

**Figure 3.** Validation of friction at different engine strokes (in N).

### 2.10. Solution Steps with Flow Chart for Computation

Figure 4 shows the flow chart for computation of this analysis, which gives the idea of the algorithm implemented for computation. As per the flow chart, the computation started at the initial crank location ($\theta = 0$). The input parameters such as minimum film thickness, average velocity, ring geometry, roughness parameters, and thermal boundary inputs are defined for that crank location. The gas pressure force and ring elastic force are calculated for that particular initial crank location. Later, the film thickness with a 3D deformation effect is estimated. Next, the elastohydrodynamic load and asperity contact force are computed. If the load convergence does not happen through film relaxation, the next iteration step is suggested. If load convergence is achieved, the computation proceeds to the next crank location, until 720° is reached. Finally, all output parameters are recorded. The model can be applicable to different cylinder diameter and thus different powers and speeds.

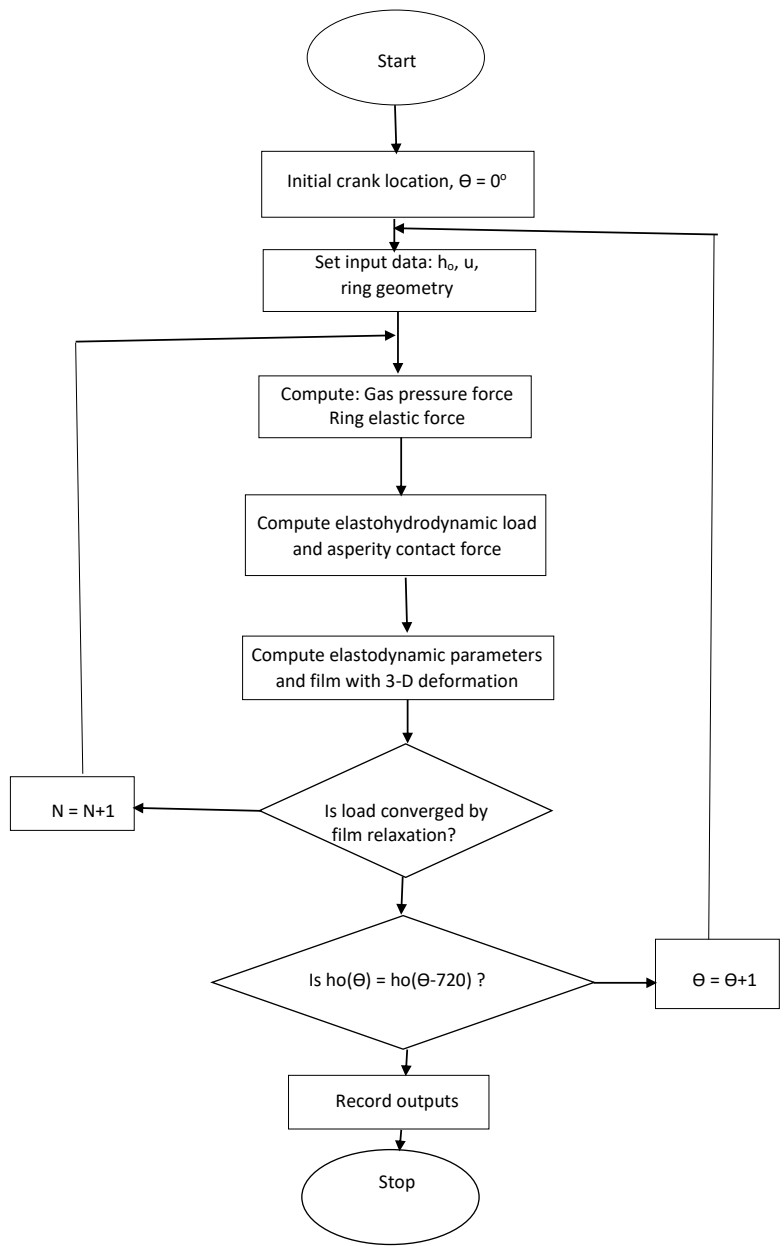

**Figure 4.** Elastodynamic and elastohydrodynamic correlation algorithm.

## 3. Results and Discussion

### 3.1. Elastodynaimcs and Elastodynamics Correlation Analysis

Figure 5 shows the bar chart comparison of the maximum value of nominal film thickness for rigid and elastic piston compression rings. At 1000 rpm engine speed, there is a 51.8% increase in the minimum film thickness in suction stroke due to consideration of elastodynamics. Whereas at the same rpm in the compression stroke, the elastic ring, compared to the rigid ring, exhibits a 47% increase in minimum film thickness. The rigid and elastic rings at 1000 rpm in the power stroke have 2.2 μm and 4.1μm film thickness, respectively, which is 50% more film in the case of the elastic ring. Finally, in the exhaust stroke at 1000 rpm, the rigid ring and elastic ring film thicknesses are 2.7 μm and 4.1 μm, respectively. Overall, at 1000 rpm, the elastic ring has more film compared to the rigid ring.

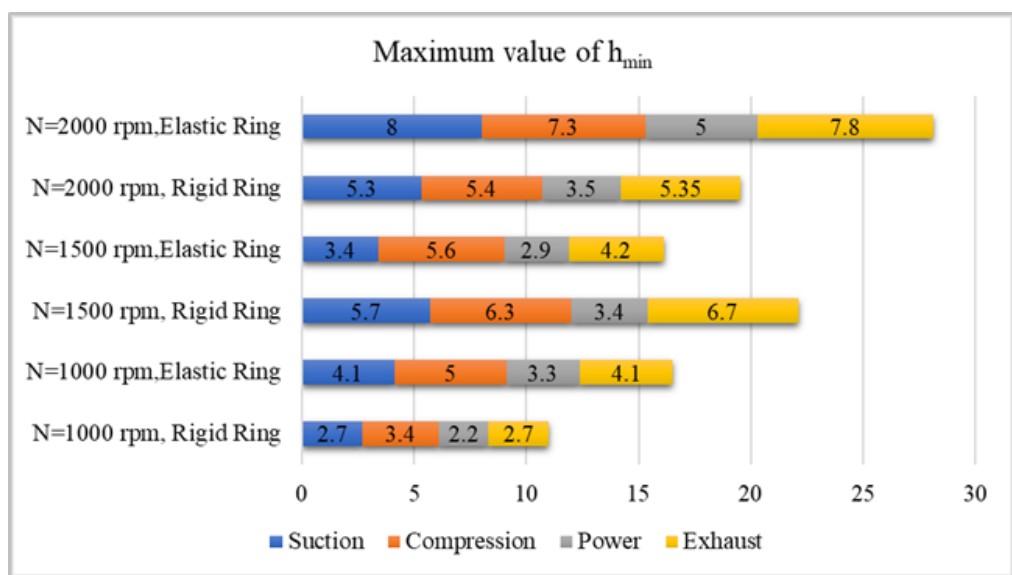

**Figure 5.** Bar chart comparison for minimum film thickness for rigid and elastic ring.

At medium speeds such as 1500 rpm, due to elastodynamic considerations, the film thickness is reduced by 40.3% in the suction stroke, while in the compression stroke at the same engine rpm, only a 9.5% reduction is observed. Whereas, in the power stroke, the elastodynamic effect reduces the minimum film by 14.7%, and in the exhaust stroke, with the same rpm, the minimum film thickness is reduced by 37.3% due to ring elastodynamics. Overall, film thickness has a reduced trend at 1500 rpm due to the elastodynamic effect. The reason is somewhat unknown.

At 2000 rpm, ring elastodynamic consideration enhances the film thickness by 51% in the suction stroke, while at the compression stroke it is enhanced by 35.2%. In the power stroke, a 42.8% enhancement is observed. Whereas in the exhaust stroke, a 38% increase in minimum film thickness is observed due to elastodynamic considerations. Overall, at 2000 rpm the trend line for film thickness increases due to the consideration of ring elastodynamics.

At an engine speed of 1500 rpm (Figure 5), the values of the nominal oil film thickness (irrespective of the engine stroke) for rigid piston rings are greater than for the flexible piston rings. For rotational speeds of 1000 rpm and 2000 rpm, the situation is reversed. Furthermore, at an engine speed of 1500 rpm, the nominal oil film thickness for a rigid ring is greater than that at 2000 rpm.

Figure 6 shows the comparison of asperity contact friction for the rigid and elastodynamic rings. At 360° crank location, boundary friction values are 80 N and 72 N for the rigid and elastodynamic ring, respectively. Considering the elastodynamic effect, a 10% decrease in asperity friction compared to a rigid ring is observed at this crank location. The reason may be due to a decrease in the real contact area of ring liner conjunction due to the elastodynamic behavior of the ring. Figure 7 presents a comparison of the maximum viscous friction for the rigid and elastodynamic rings. At the 373° crank location, friction values are 68 N and 52 N for rigid and elastic rings, respectively. Due to the elastodynamic consideration, there is a 23.53% decrease in fluid friction. This may be due to the influence of elastic ring global deformation on the film profile of the ring–liner conjunction. Tables 4–6 present the comparison of oil film, friction power loss, and lubricant oil flow, respectively, in the different engine strokes for rigid and elastic ring consideration. The tables compare the maximum value of stated parameters of suction, compression, power, and exhaust stroke of a four-stroke single cylinder engine at three different speeds: 1000 rpm, 1500 rpm, and 2000 rpm.

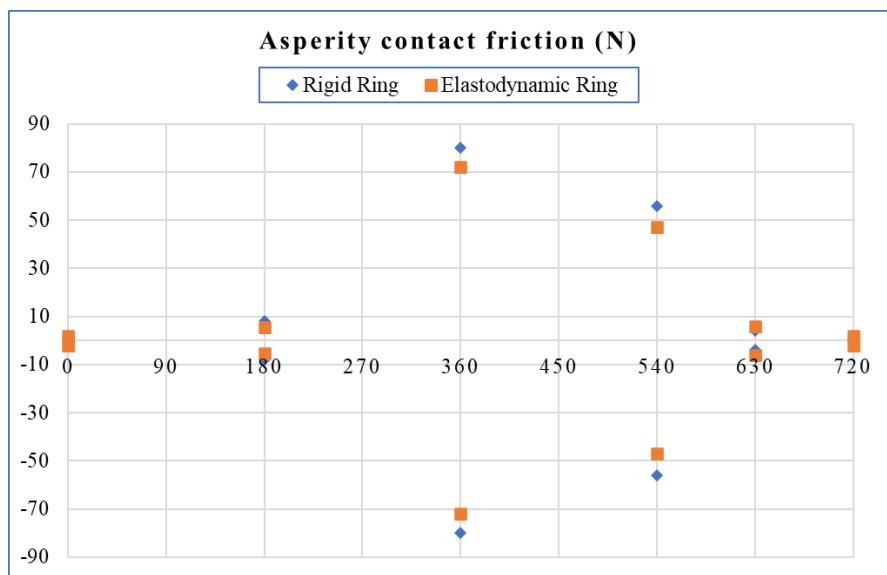

**Figure 6.** Asperity contact friction comparison.

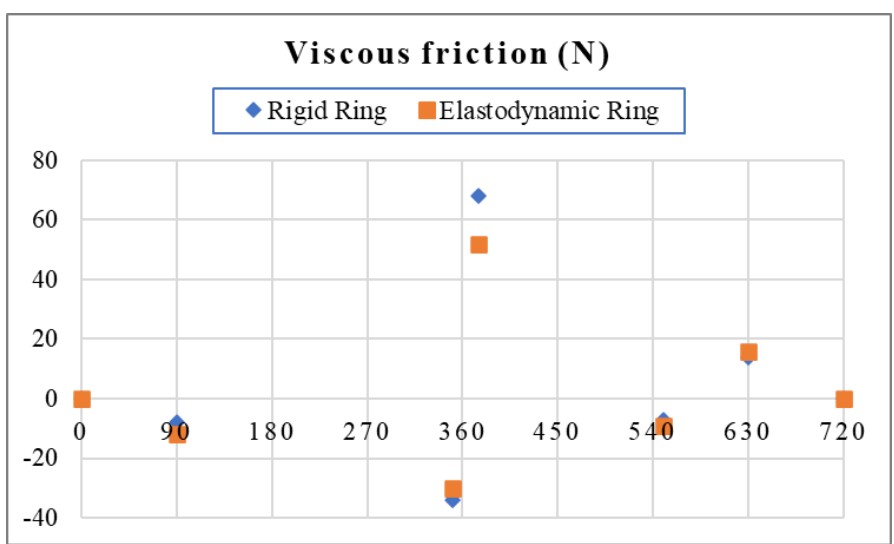

**Figure 7.** Maximum viscous friction.

**Table 4.** Film thickness comparison.

|  | N = 1000 rpm, Rigid Ring | N = 1000 rpm, Elastic Ring | N = 1500 rpm, Rigid Ring | N = 1500 rpm, Elastic Ring | N = 2000 rpm, Rigid Ring | N = 2000 rpm, Elastic Ring |
|---|---|---|---|---|---|---|
| Suction | 2.7 | 4.1 | 5.7 | 3.4 | 5.3 | 8 |
| Compression | 3.4 | 5 | 6.3 | 5.6 | 5.4 | 7.3 |
| Power | 2.2 | 3.3 | 3.4 | 2.9 | 3.5 | 5 |
| Exhaust | 2.7 | 4.1 | 6.7 | 4.2 | 5.35 | 7.8 |

Figure 8 shows the comparative bar chart for the rigid and elastic rings and related maximum frictional power loss for various strokes. At 1000 rpm, for the suction stroke, a 71% increase in friction power loss is observed, while at the same speed, the elastodynamic consideration enhances friction power loss; it is enhanced by 28.6% in the compression

stroke. The power stroke ring elastodynamics account for a 21.2% enhancement in frictional power loss, whereas in the exhaust stroke at 1000 rpm, the power loss increases by 50% due to inclusion of elastodynamic ring properties. Overall, at 1000 rpm the ring elastodynamic effect causes power loss in the ring–liner conjunction. At the medium speed of 1500 rpm in the suction stroke, compared to the rigid ring, the elastic ring friction power loss is 39.5% more, whereas in the compression stroke there is a 22.72% increase in power loss compared to the rigid ring. However, in the power stroke, a 14.3% power loss is observed due to replacing the elastic ring with a rigid ring at 1500 rpm. Whereas in the exhaust stroke, the friction loss increases by 26.5%.

**Table 5.** Friction power loss comparison.

|  | N = 1000 rpm, Rigid Ring | N = 1000 rpm, Elastic Ring | N = 1500 rpm, Rigid Ring | N = 1500 rpm, Elastic Ring | N = 2000 rpm, Rigid Ring | N = 2000 rpm, Elastic Ring |
|---|---|---|---|---|---|---|
| Suction | 38 | 65 | 86 | 120 | 135 | 180 |
| Compression | 42 | 54 | 82.3 | 101 | 143 | 160 |
| Power | 80 | 250 | 340 | 280 | 240 | 330 |
| Exhaust | 42 | 63 | 83 | 105 | 134 | 160 |

**Table 6.** Lubricant flow comparison.

|  | N = 1000 rpm, Rigid Ring | N = 1000 rpm, Elastic Ring | N = 1500 rpm, Rigid Ring | N = 1500 rpm, Elastic Ring | N = 2000 rpm, Rigid Ring | N = 2000 rpm, Elastic Ring |
|---|---|---|---|---|---|---|
| Suction | 0.051 | 0.06 | 0.116 | 0.095 | 0.13 | 0.16 |
| Compression | 0.046 | 0.058 | 0.11 | 0.085 | 0.131 | 0.145 |
| Power | 0.032 | 0.038 | 0.065 | 0.06 | 0.09 | 0.1 |
| Exhaust | 0.051 | 0.06 | 0.115 | 0.083 | 0.128 | 0.16 |

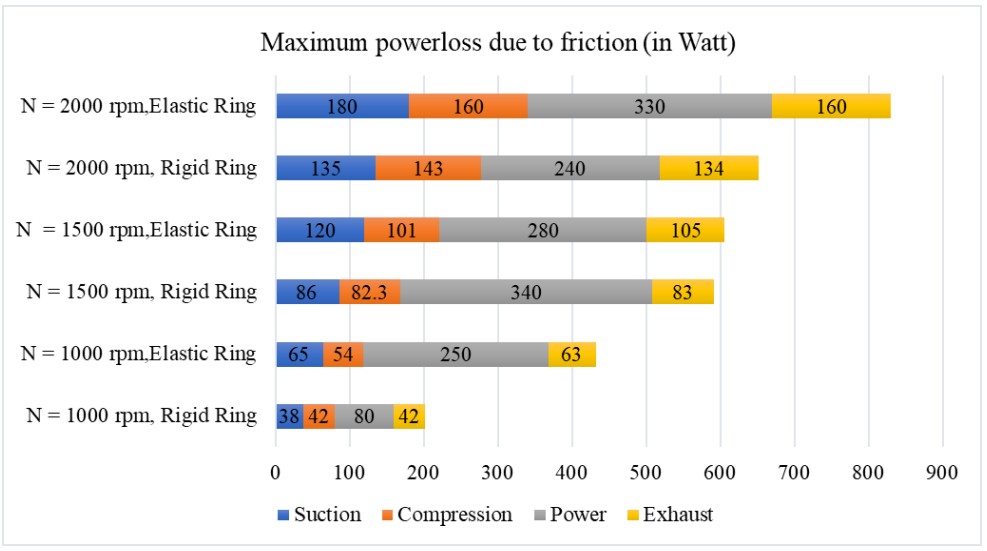

**Figure 8.** Bar chart comparison for friction power loss for rigid and elastic rings.

At an engine speed of 2000 rpm, due to elastic dynamic consideration, a 33.4% increase in friction power loss is observed in the suction stroke, while at the same speed

in the compression stroke, there is an 11.88% enhancement in friction power due to ring elastodynamic consideration. Whereas in the power stroke, a 37.5% increase in power loss is observed. Finally, the exhaust stroke accounts for a 19.4% increase in power loss due to the elastodynamic consideration of the ring. Overall, at 2000 rpm the trend line for friction power loss increases due to the consideration of ring elastodynamics.

Figure 9 shows the bar chart for a comparison of lubricant oil flow at various speeds and strokes between a rigid ring and an elastodynamic ring. At a lower engine speed of 1000 rpm, for the suction stroke, a 17.64% increase in oil flow is observed, while at the same speed, the elastodynamic consideration enhances the oil flow loss by 26% in the compression stroke. The power stroke ring elastodynamics account for an 18.75% increase in oil flow, whereas in the exhaust stroke at 1000 rpm, the oil flow increases by 17.6% due to inclusion of the elastodynamic ring properties. Overall, at 1000 rpm, the ring elastodynamic effect adds oil flow in the ring–liner conjunction.

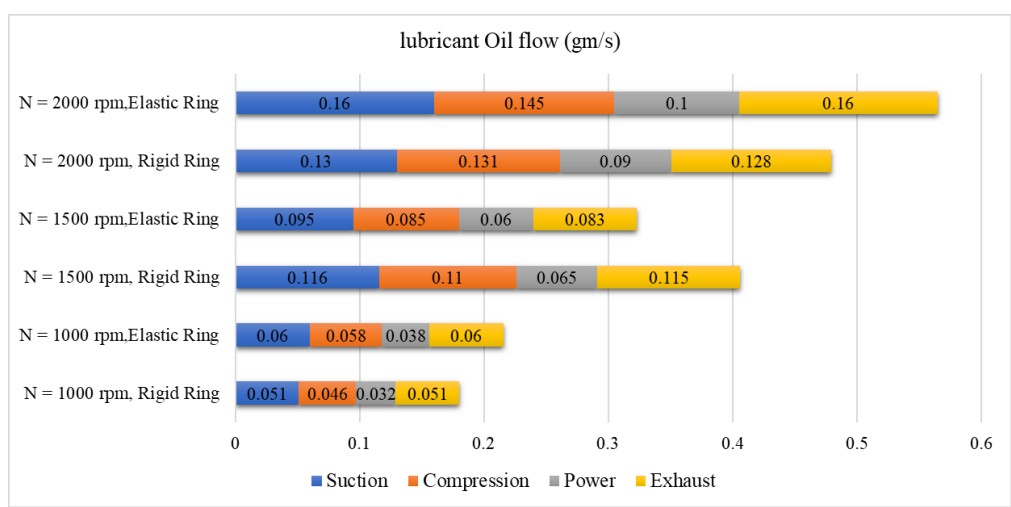

**Figure 9.** Bar chart comparison for lubricant oil flow (gm/s) for rigid and elastic rings.

At the medium speed of 1500 rpm in the suction stroke, compared to the rigid ring, the elastic ring oil flow is 19.1% less, whereas in the compression stroke there is a 22.72% decrease in oil flow compared to rigid ring. However, in the power stroke, a 7.7% lower oil flow is observed due to replacing the elastic ring with a rigid ring at 1500 rpm. Whereas in the exhaust stroke, such an oil flow decreases by 26.1%.

At an engine speed of 2000 rpm, due to the elastic dynamic consideration, a 23.07% increase in oil flow is observed in the suction stroke, while at the same speed in the compression stroke, there is a 10.7% enhancement in oil flow due to ring elastodynamic consideration. Whereas in the power stroke, an 11.11% increase in oil flow is observed. Finally, the exhaust stroke accounts for a 25% increase in oil flow due to the elastodynamic consideration of the ring. Overall, at 2000 rpm the trend line for oil flow increases due to the consideration of ring elastodynamics.

*3.2. Finite Element Analysis of Elastic Ring Subjected to Elastodynamics*

The finite element simulation was conducted using ANSYS, in which the SOLID-QUAD 4NODE 182 type elements are auto-selected in the ANSYS workbench. The selected material for the core and coated surface of the ring is given in Table 7. Nikasil is the abbreviation for nickel silicon carbide. This silicon carbide is an extremely hard ceramic (much harder than steel) and capable of dissolving in nickel. Such coating provision is conducive for simultaneous sealing and sliding actions, which are prevalent in the case of ring–liner contact.

**Table 7.** Material details of coated ring for FEA.

| Component | Material | E | $\upsilon$ |
|-----------|----------|---|---|
| Ring core | Steel | 205 GPa | 0.3 |
| Coating | Nikasil | 110 GPa | 0.2 |

Figure 10 shows the layout drawing of the incomplete circular ring drawn for simulation. Once this design was ready, the meshing of the core as well as the coating of the ring was carried out to develop the whole surface into several convenient elements. The QUADRILATERAL solid elements thus generated are connected through nodes. For proper meshing, the model is set to smart size control on picked lines. The core area was set under the element attribute with reference to material along with element type. Core meshing, as well as coating area meshing, was carried out in a free quad shape. For one crank location, the free-body diagram of the ring is given in Figure 11.

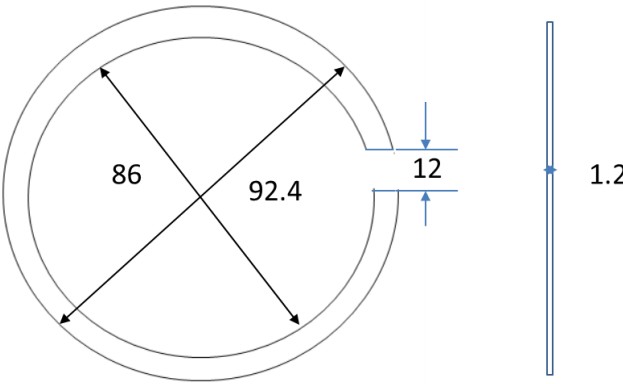

**Figure 10.** Layout drawing of an incomplete piston compression ring.

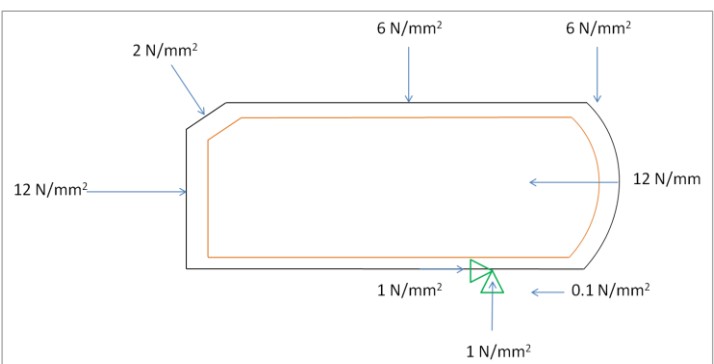

**Figure 11.** Pressure force considered at a different face of the ring for FEA analysis.

The groove friction was considered negligible in this analysis as it is insignificant compared to other active forces. The ring is considered to be hinged at the support and the piston block is an upward motion in the compression stroke. Figure 12a shows the 3D deformation of the ring due to the application of this force. Figure 12b shows the von-Mises stress, with a maximum value of 0.414 N/mm$^2$.

Figure 12c shows the deformation in the interface with 0.513 µm as maximum deformation of the coating surface. Whereas the corresponding von-Mises stress is 0.414 N/mm$^2$, as given in Figure 12d.

Figure 13a shows the von-Mises strain of the elastodynamic ring, and the maximum strain is $6.74 \times 10^{-07}$ at the point of contact of the ring and groove land. Whereas Figure 13b shows the shear stress distribution on the ring, where the maximum value was found to be 0.11 N/mm$^2$.

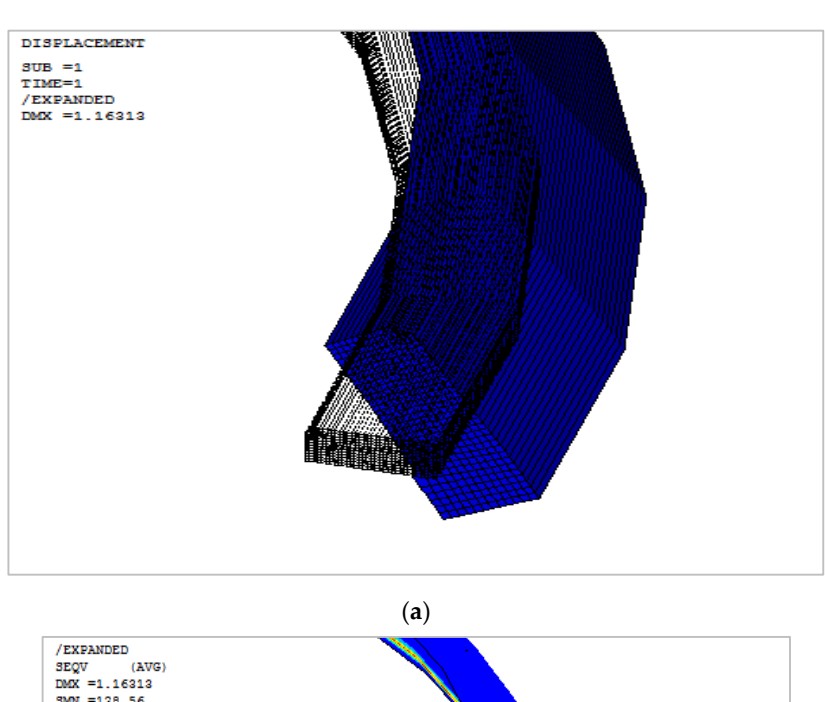

(**a**)

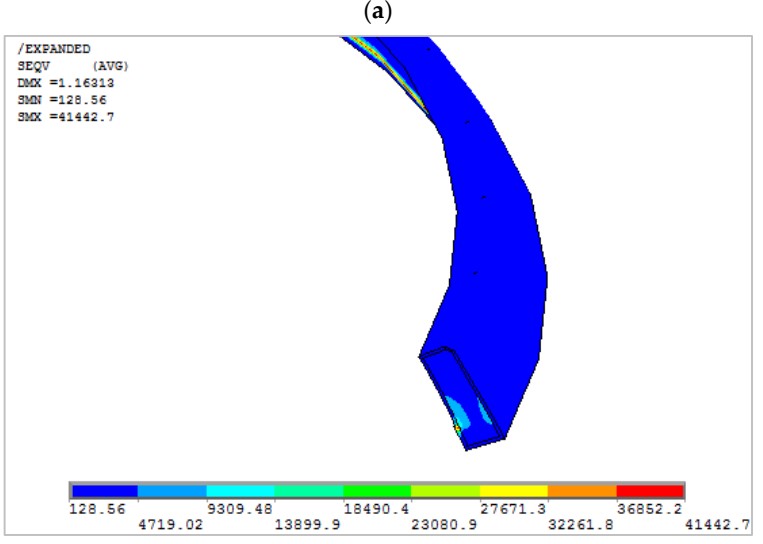

(**b**)

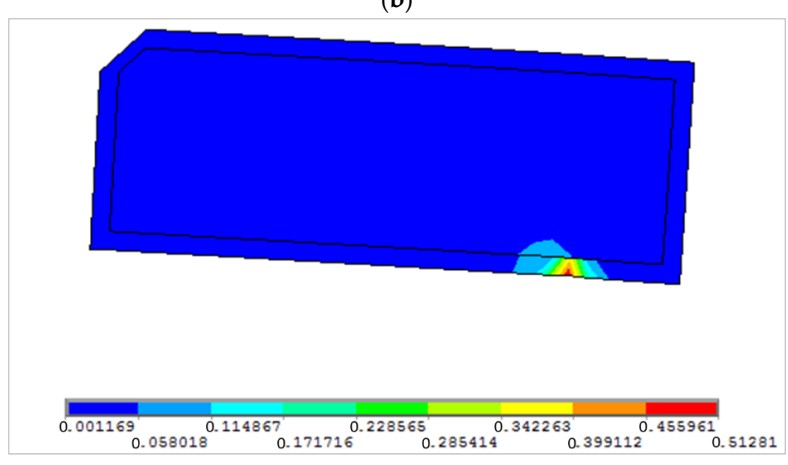

(**c**)

**Figure 12.** *Cont.*

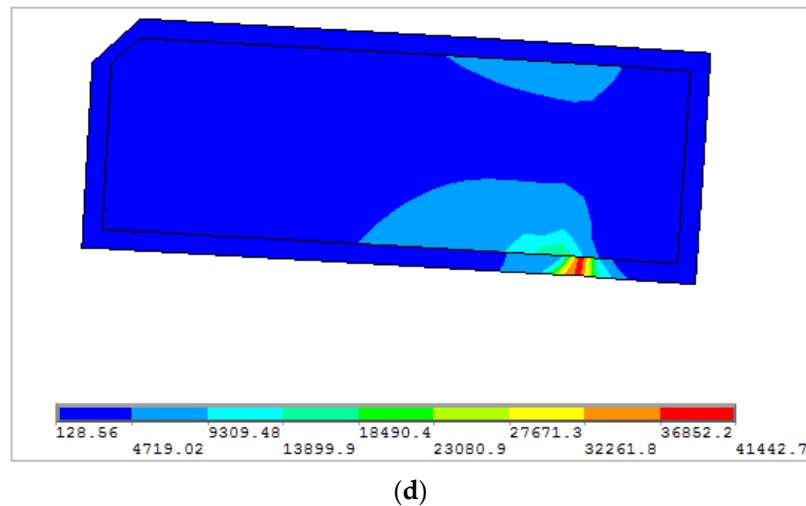

(**d**)

**Figure 12.** Elastodynamics of the compression ring through finite element analysis (**a**) 3D deformed shape (**b**) von-Mises stress, (**c**) interface deformation in μm, (**d**) von-Mises stress sectional distribution.

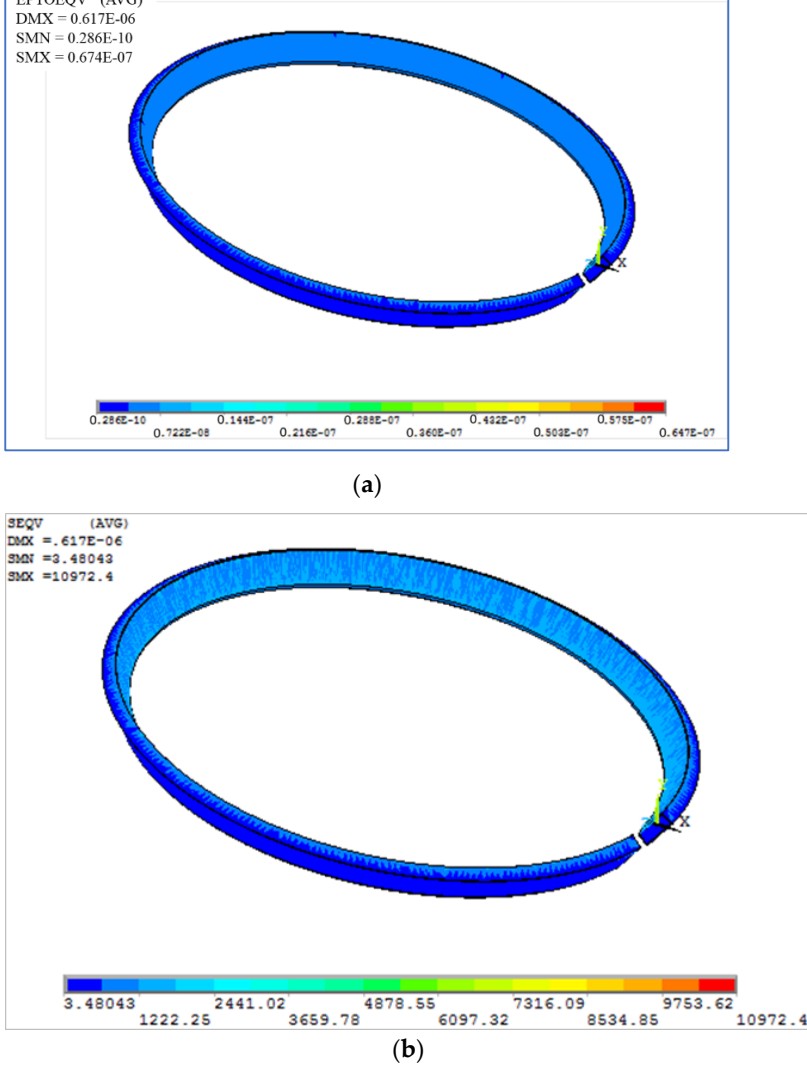

**Figure 13.** Elastodynamics of compression ring: (**a**) von-Mises strain, (**b**) shear stress of total deformed ring.

## 4. Conclusions

This study discussed a novel approach of correlating the compression ring elastodynamics and ring-line elastohydrodynamics. The 3D deformation of the ring was interrelated to the in-plane and out-of-plane dynamic forces through lubricant film thickness. The following conclusions are made from the research:

- The maximum increase in film thickness was observed to be 51.8% in the suction stroke, while it was 47% in the compression stroke, 50% in the power stroke and 52% in the exhaust stroke;
- The asperity contact force is reduced by 10% and 23.53% due to elastodynamic considerations at top and bottom dead center, respectively;
- The elastic ring shows reduced asperity friction loss compared to rigid ring at the crank location of 360° (at TDC);
- The highest viscous friction is reduced by 23.53% due to consideration of ring elastodynamics at 373° crank location;
- More friction power loss in the suction stroke compared to other strokes was observed, with a maximum increase of 72%;
- Frictional power loss remarkably decreases at 1500 rpm in the power stroke due to consideration of elastodynamics;
- When the speed of rotation increases, the lubricant oil flow to the conjunction increases for both the rigid and elastic rings;
- At 1500 rpm, both film thickness and oil flow decrease due to consideration of elastodynamics;
- Oil flow increases by a maximum of up to 26% during the compression stroke at 1000 rpm, while its lowest is 7.7% in the power stroke at 1500 rpm.

The sustainability of elastohydrodynamic lubrication of piston compression ring–liner contacts largely depends on the elastodynamic behavior of the ring. The limitation of this current work is that it does not include the ring–groove land interaction in the dynamic analysis. In addition, the secondary tilting and its effect on ring twist is not discussed in this analysis. The difficult operating conditions of the piston–piston ring–cylinder contact in an internal combustion engine, including the sealing ring, result not only from high temperatures and high pressures in the combustion chamber, but also from the interaction of hard mineral particles that enter the cylinder with the intake air, and this is scope for future work of this research.

**Author Contributions:** Conceptualization, P.C.M.; Methodology, S.B.; Software, S.B.; Validation, P.C.M.; Formal analysis, S.B.; Investigation, P.C.M.; Supervision, P.C.M. All authors have read and agreed to the published version of the manuscript.

**Funding:** All authors are thankful to MDPI for considering full waiver of the APC for publication of this manuscript.

**Data Availability Statement:** All data are part of the manuscript. No external data required.

**Conflicts of Interest:** The authors declared no conflict of interest.

## List of Symbols

| | | |
|---|---|---|
| $A$ | Cross sectional area of the ring | mm$^2$ |
| $A_{a/e}$ | Asperity contact area | mm$^2$ |
| $C_a$ | Coefficient used to calculate asperity contact pressure | |
| $D$ | Bore nominal diameter | mm |
| $E$ | Elastic modulus of the ring material | GPa |
| $F_T$ | Tangential shear force | N |
| $F_{\frac{5}{2}}$ | Greenwood and Trip statistical function | |
| $f_b$ | Force acting on the boundary interaction | N |
| $f_v$ | Force component due to viscous action | N |

| | | |
|---|---|---|
| $F_{in\_plane}$ | Net in-plane force | N |
| $F_{out\_of\_plane}$ | Net out-of-plane force | N |
| $F_e$ | Ring elastic force | N |
| $F_g$ | Applied gas pressure force | N |
| $H$ | Distance between oil ring bottom land and piston pin center | mm |
| $h_1$ | Distance between oil ring bottom land and piston CG | mm |
| $h_2$ | Distance between pin center and piston CG | mm |
| $h_0(\theta_c)$ | Minimum gap between out-of-round bore and conformed ring | mm |
| $h_T$ | Total film thickness | μm |
| $h$ | Nominal film thickness | μm |
| $h_x$ | Film thickness in sliding direction | μm |
| $h_y$ | Film thickness in side-leak direction | μm |
| $I$ | Moment of inertia of the ring | mm$^4$ |
| $m$ | Mass of the ring per unit length | gm |
| $p$ | Pressure | N/mm$^2$ |
| $p_{atm}$ | Atmospheric pressure | N/mm$^2$ |
| $Q_n$ | General force function | |
| $R$ | Ring nominal crown radius | mm |
| $S_{i,j}$ | Ring shape function | μm |
| $t$ | Time | s |
| $W_h$ | Lubricant reaction | N |
| $W_a$ | Asperity contact load | N |
| Greek Symbols | | |
| $Z$ | Coefficient for asperity contact calculation | |
| $\alpha_0$ | Piezo-viscous coefficient | Pa$^{-1}$ |
| $\beta_0$ | Thermo-viscous coefficient | |
| $\beta, k$ | Oil dependent fitting parameters | |
| $\mu$ | Shear dependent viscosity | Pa.s |
| $\mu_1$ | Low shear rate viscosity | Pa.s |
| $\mu_2$ | High shear rate viscosity | Pa.s |
| $\gamma^*$ | Shear rate | s$^{-1}$ |
| $\Theta$ | Temperature of the lubricant | °K |
| $\Theta_0$ | Initial temperature of the lubricant | °K |
| $\eta$ or $\eta_0$ | Viscosity or reference viscosity | Pa.s |
| $\tau$ or $\tau_0$ | Shear rate or reference shear rate | N/mm$^2$ |
| $\rho_0$ or $\rho$ | Reference density or density | kg/m$^3$ |
| $\xi$ | Pressure coefficient of boundary friction | |
| $\varphi$ | Ring circumferential location | degree |
| $\Delta_{i,j}$ | Ring global deformation | μm |
| $\delta_{i,j}$ | Ring local deformation | μm |
| $\Omega$ | Parameter to calculate asperity contact pressure | |
| $\lambda$ | Parameter to calculate asperity contact pressure | |

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
