# Peer review of "Piston Compression Ring Elastodynamics and Ring–Liner Elastohydrodynamic Lubrication Correlation Analysis"

_lubricants, doi:10.3390/lubricants10120356_

Round 1
Reviewer 1 Report
Uwagi w załączniku

Author Response
Dear Sir,
thank you for the valuable suggestions. We corrected the manuscript according to the suggestions and highlighted the changes. Please find the response to the review comments as:
- There is no clearly formulated objective of the work
Response# Please find the objective is now upgraded
- Results not have not been verified
Response# Please find the validation, which was offered earlier is now given.
- The model developed lacks assumptions
Response# Please find the assumptions are now clearly given in a subsection
- There is also a lack of a calculation algorithm, where the input data for the calculations should be given.
Response# Please find the calculation algorithm is now added
- Text and figures carelessly prepared.
Response# Please find the text now organized.
- The conclusions should be more consistent with the content of the work and the stated objective.
Response# Please find the conclusion is now upgraded.
- The introduction is dominated by literature from years below 2018
Response# Please find now the literature is updated with new suggestions
- The most recent literature in the field, especially from the last 5 years, has not been fully exploited.
Response# Please find now it is updated.
- The analysis of the literature should be expanded to include other items covering the problem at hand, such as:
https://doi.org/10.3390/ma14226839
https://doi.org/10.4271/2020-01-2227
https://doi.org/10.19206/CE-2016-403
Response# Please find these literatures are added and the reference are now updated.
- The authors should show more of the difficult operating conditions of the P-PR-C association of an internal combustion engine, including the sealing ring, which result not only from high temperatures and high pressures in the combustion chamber, but also from the interaction of hard mineral particles that enter the cylinders with the intake air. There is an argument for undertaking this type of research. The following publications may be helpful here:
https://doi.org/10.3390/en15031182
https://doi.org/10.3390/en15134815
https://doi.org/10.1504/IJSURFSE.2020.105891
Response# Please find now these references are added. We feel this comment is extremely useful to implement. Due to limitation of appropriate research facility, it is a constraint for us to go beyond 2000 rpm. However, we mention it in the conclusion for our future research plan. Thank you.
- What medium is the density of ρ0=1800 kg/m3 - line 274.
Response# Please find the typo mistake is now corrected it is 883.8 kg/m3.
- Why engine speeds of 1000, 1500, 2000 rpm were used for the calculations.
Response# It is for your kind information that the engine speed is maximum of 2500 rpm. So, we decided to keep it on the safe range.
- At an engine speed of 1500 rpm (Figure 3), the values of the nominal oil film thickness (irrespective of the engine stroke) for rigid piston rings is greater than for and flexible piston rings. For rotational speeds of 1000 rpm and 2000 rpm, the situation is reversed. Furthermore, at an engine speed of 1500 rpm, the nominal oil film thickness for a rigid ring is greater than at 2000 rpm. Please analyses and explain accordingly and include in the text of the manuscript.
Response# Please find this is implemented.
- The ring shown in Figure 1 is not circular - the end of the ring is a straight line. Is this the way it should be?
Response# It was zoomed. To correct we replaced with a better figure 1.
- The piston ring shown in Figure 8 has an inner diameter of 86 mm and an outer diameter of 92 mm. The thickness of the ring is 3.2 mm. If we add the two ring thicknesses to the outer diameter we get 92.4 mm, not 92 mm. What was the cylinder diameter D of the engine adopted for the simulation?
Response# We are sorry for this typo error. Please find the value is now corrected.
- The following information is given in the list of symbols: Weight of the ring in g. Or the mass of the ring in gm.
Response# Please find this mistake is now rectified
- The expression "mg" is not mass as it says "Weight of the ring per unit length". The unit of this expression may also not be "g" but "N".
Response# The author feel sorry for this error. Now it is corrected
- in the list of symbols - clarify the difference between the unit for oil viscosity "Pas" and "PaS". in the list of symbols, the following information is given: If it is the quotient of two densities, there is no unit for this expression. Furthermore, the unit of density is kg/m3 , not Kg/m3.
Response# It is for your kind information that this is not quotient. Please find the typo is now corrected.
- In formula 13 and 14 the expression (p - Patm) occurs. What is meant by the symbol Patm. The list of symbols lacks this designation. If it is pressure, why is the capital letter “P” used.
Response# It is pressure for your kind information. Please find now it is corrected.
- In the lubrication model developed, I have not noticed a quantity that takes into account the speed of the piston and ring movement in the formation of the oil film between the ring and the engine cylinder.
Response# Please refer to the Eq.(18) to find the uav is the average velocity of entrainment.
- Is the developed model applicable to an engine with specific parameters or can it be used for an engine with different cylinder diameters D and thus different power and speed. There are no given assumptions for the developed model in the paper.
Response# The model can be applicable to different cylinder diameter thus different power and speed. Please find it is mentioned in the manuscript.
- The temperature of the engine has a significant effect on changes in oil viscosity, especially when operating at low temperature conditions - below 0o In what range of temperature changes (oil viscosity changes) is the developed model useful.
Response# It is for your kind information that the friction led to a temperature change of 90oC in the calculation.
- the developed model has not been verified. The authors do not refer to this problem. Please provide an opinion in this regard.
Response# Please find it is now verified.
- The conclusions should be reworded to be more consistent with the content of the paper and relevant to the problem at hand. The conclusions have little reference to the model developed and the research results obtained and the purpose of the work.
Response# Please find the conclusion is now upgraded.
We are thankful to our learned colleague for so many valuable corrections. Due to which the manuscript is improved to a significant level. We appreciate the effort, and open for further correction if raised during further process of review.
Reviewer 2 Report
1) Include key results in abstract.
2) The results are merely described. The authors are encouraged to include a discussion section and critically discuss the observations from this investigation with existing literature.
3) Suggestions of literature:
a) book, Tribology for Engineers:A Practical Guide. Elsevier, 2011, ISBN: 9780081014912
b) book, Progress in Green Tribology, DE Gruyter, 2017, ISBN: 9783110372724
c) book, Mechanical and Industrial Engineering, Springer, 2022 ISBN: 978-3-030-90486-9
d) book, Modern Mechanical Engineering, Springer, 2014, ISBN978-3-642-45175-1
and articles of reputed international journals.
4) Improve the conclusions
Author Response
Dear Sir,
thank you for the valuable suggestions. We corrected the manuscript according to the suggestions and highlighted the changes. Please find the response to the review comments as:
- Include key results in abstract.
Response# Please find the key results are further reported and highlighted.
- The results are merely described. The authors are encouraged to include a discussion section and critically discuss the observations from this investigation with existing literature.
Response# Please find the results are discussed further.
- Suggestions of literature:
- a) book, Tribology for Engineers:A Practical Guide. Elsevier, 2011, ISBN: 9780081014912
- b) book, Progress in Green Tribology, DE Gruyter, 2017, ISBN: 9783110372724
- c) book, Mechanical and Industrial Engineering, Springer, 2022 ISBN: 978-3-030-90486-9
- d) book, Modern Mechanical Engineering, Springer, 2014, ISBN978-3-642-45175-1
and articles of reputed international journals.
Response# Please find theses literatures are referred and cited
- Improve the conclusions
Response# Please find the conclusion is now improved.
We are thankful to our learned colleague for so many valuable corrections. Due to which the manuscript is improved to a significant level. We appreciate the effort and are open to further correction if raised during the further process of review.
Round 2
Reviewer 2 Report
No comments
Author Response
We are thankful to our learned colleague for so many valuable corrections. Due to this the manuscript is improved to a significant level. We appreciate the effort and are open to further correction if raised during the further process of review.
- Include key results in the abstract.
Response# Please find the key results that are further reported and highlighted.
- The results are merely described. The authors are encouraged to include a discussion section and critically discuss the observations from this investigation with existing literature.
Response# Please find the results discussed further.
- Suggestions of literature:
- a) book, Tribology for Engineers:A Practical Guide. Elsevier, 2011, ISBN: 9780081014912
- b) book, Progress in Green Tribology, DE Gruyter, 2017, ISBN: 9783110372724
- c) book, Mechanical and Industrial Engineering, Springer, 2022 ISBN: 978-3-030-90486-9
- d) book, Modern Mechanical Engineering, Springer, 2014, ISBN978-3-642-45175-1
and articles of reputed international journals.
Response# Please find theses literatures are referred and cited
- Improve the conclusions
Response# Please find the conclusion is now improved.